# Chemical Modification of Cytochrome C for Acid-Responsive Intracellular Apoptotic Protein Delivery for Cancer Eradication

**DOI:** 10.3390/pharmaceutics16010071

**Published:** 2024-01-04

**Authors:** Bo Tang, Kwai Man Lau, Yunxin Zhu, Chihao Shao, Wai-Ting Wong, Larry M. C. Chow, Clarence T. T. Wong

**Affiliations:** State Key Laboratory of Chemical Biology and Drug Discovery, Department of Applied Biology and Chemical Technology, Hong Kong Polytechnic University, Kowloon, Hong Kong, China; bo-gary.tang@connect.polyu.hk (B.T.); kwai-man-wendy.lau@connect.polyu.hk (K.M.L.); yunxin-avery.zhu@connect.polyu.hk (Y.Z.); chi-hao1.shao@connect.polyu.hk (C.S.); wai-ting.wong@polyu.edu.hk (W.-T.W.)

**Keywords:** cytochrome C, cell penetrating peptide, acid-responsive, protein modification

## Abstract

Delivering bioactive proteins into cells without carriers presents significant challenges in biomedical applications due to limited cell membrane permeability and the need for targeted delivery. Here, we introduce a novel carrier-free method that addresses these challenges by chemically modifying proteins with an acid-responsive cell-penetrating peptide (CPP) for selective intracellular delivery within tumours. Cytochrome C, a protein known for inducing apoptosis, served as a model for intracellular delivery of therapeutic proteins for cancer treatment. The CPP was protected with 2,3-dimethyl maleic anhydride (DMA) and chemically conjugated onto the protein surface, creating an acid-responsive protein delivery system. In the acidic tumour microenvironment, DMA deprotects and exposes the positively charged CPP, enabling membrane penetration. Both in vitro and in vivo assays validated the pH-dependent shielding mechanism, demonstrating the modified cytochrome C could induce apoptosis in cancer cells in a pH-selective manner. These findings provide a promising new approach for carrier-free and tumour-targeted intracellular delivery of therapeutic proteins for a wide range of potential applications.

## 1. Introduction

Intracellular protein delivery offers significant potential in novel therapeutics, encompassing a wide range of applications from directly modulating cellular pathways to treating genetic disorders [1,2,3]. This approach represents a frontier in medicine, with the capacity to transform our understanding and treatment of various complex diseases by precisely targeting and manipulating the fundamental processes within cells, thereby opening new avenues for ground-breaking medical interventions [4]. However, intracellular protein delivery faces significant challenges that have limited its full potential. The cell membrane presents a significant barrier to protein entry. Moreover, achieving targeted delivery while minimising off-target effects remains a hurdle. Therefore, different types of nanocarriers have been developed to improve the delivery process. To date, researchers have developed two strategies to deliver nanocarrier-loaded with bioactive proteins across the cell membrane into tumours. The first approach involves attaching tumour-specific receptor-targeting groups to the carriers, initiating receptor-mediated endocytosis [5]. This method is highly specific for targeting cancer cells, yet it often results in the cargo being transported to endosomes and lysosomes, where most proteinaceous cargos are trapped, denatured, and degraded. The second approach employs cell-penetrating peptides (CPPs) attached onto the surface of nanocarriers to facilitate direct penetration of the membrane, delivering the protein cargo into the cytoplasm or even the nucleus [6]. Cell-penetrating peptides (CPPs) are short peptides (5–30 amino acids) that possess the unique ability to translocate across the cell membrane while preserving membrane integrity, exhibiting minimal invasiveness, and demonstrating low cytotoxicity [7,8,9]. The versatility of CPPs allows them to transport a range of cargo molecules, including peptides, proteins, nucleic acids, small molecules, and nanoparticles [10,11,12,13,14,15]. This capability has spearheaded numerous preclinical and clinical studies [16,17,18].

To add additional control to cargo release, acid-responsive mechanisms were introduced. Acid-responsive drug delivery methods, crucial for targeting specific tissues like tumours, are divided into different strategies, such as physical dissociation in acidic conditions and carrier swelling at pH-sensitive micelles that swell to release their payload, all leveraging nanomaterials for precise, targeted delivery [19]. Chemical bond cleavage in nanoparticles can also result in nanoparticle dissociation or promote payload release by utilising acid-labile bonds in copolymers for content release [20]. 2,3-dimethyl maleic anhydride (DMA) was developed to reduce the off-target delivery of CPP-linked cargos by reversibly masking the amino groups on lysine residues on the CPP via a nucleophilic attack by the amine nitrogen, resulting in the generation of a cyclic imide intermediate and the generation of a negatively charged carboxylate group to prevent cell penetrating [21,22,23,24,25,26]. The imide bond formed between DMA and the amine group is acid-labile, which can be cleaved under slightly acidic conditions of pH < 6.8, which is similar to the tumour microenvironment. Consequently, the original amine group is regenerated, allowing the positively charged CPP to be exposed and regain its cell-penetrating ability. Such a DMA-CPP approach has been used in many nanoparticle and material designs [27,28,29,30,31]. However, nanoparticle carriers frequently face challenges such as complex characterisation processes, low efficiency, intricate release mechanisms, endosomal entrapment, and significant batch-to-batch variability. These obstacles make it challenging to translate these carriers into effective therapeutic applications.

On the other hand, the molecular delivery approach for direct intracellular protein delivery presents distinct advantages, foremost being its precision and accuracy in dosage control. Without nanocarrier, it also eliminates the complexities and unpredictability often associated with release mechanisms in carrier-based systems. Additionally, the carrier-free approach significantly simplifies the production process, effectively eliminating the batch-to-batch variations that are a common challenge in carrier-based drug delivery systems [32]. This results in a more consistent and reliable therapeutic product. However, molecular tumour-selective protein delivery systems remained rare and difficult to achieve. The conventional method of delivering native protein with CPP involves coexpressing a CPP sequence at one of the protein’s termini to confer its cell-penetrating ability [33,34]. This approach, however, faces challenges such as off-target penetration and alterations to the bioactive structure and folding of proteins due to CPP fusion, thereby potentially compromising production and therapeutic efficacy.

Herein, we have utilised the acid-responsive properties of the aforementioned DMA-CPP mechanism, traditionally used in nanosystems, and innovatively applied it to a native protein, cytochrome C, through chemical modification. This process involves the conjugation of a chemically synthesised CPP equipped with an acid-responsive DMA group. The resulting conjugate, **DMA.CPP-CytC**, is designed to penetrate the cytoplasm of tumour cells selectively in slightly acidic conditions, thereby inducing an apoptotic effect, as illustrated in Figure 1. The integration of the DMA-CPP system with cytochrome C represents a novel approach, aiming to enhance the targeted delivery and efficacy of the bioactive protein within cancer cells, leveraging the acidic tumour microenvironment to trigger a therapeutic response.

To this end, we developed a chemical approach that employed a bifunctional linker to perform a robust three-step reaction for modifying the lysine residues on the native cytochrome C surface with DMA-protected CPP. The addition of multiple CPPs to the protein surface is expected to improve transmembrane efficiency, providing a more effective means of intracellular protein delivery. Furthermore, the DMA on the peptide provides an extra layer of selectivity that is preferentially internalised at the acidic microenvironment of the tumour site, making it a potential candidate for cancer therapy.

## 2. Materials and Methods

### 2.1. Preparation of Cell Penetrating Peptide (CPP)

The cell-penetrating peptide BP16 (Fmoc-GCKKLFKKILKKL-CONH_2_) [9], to which cysteine was added at the N-terminus for conjugation purposes, was synthesised using a 9-fluorenylmethoxycarbonyl (Fmoc) solid-phase peptide synthesis protocol with commercially available N-α-Fmoc-protected amino acids. The CEM Liberty Blue Automated Microwave Peptide Synthesiser was used for synthesis, with rink amide resin as the solid support. The Fmoc protecting group was removed using a solution of 20% piperidine in DMF, and 1-[bis(dimethylamino)methylene]-1H-1,2,3-triazolo-[4,5-b]pyridinium 3-oxide hexafluorophosphate (HATU) was used as the carboxyl group activating agent. Each coupling reaction was carried out using an excess of the Fmoc-protected amino acid (4 equiv.), HATU (4 equiv.), and *N,N*-diisopropylethylamine (DIPEA) (8 equiv.) in DMF at room temperature. N-terminal acetylation was achieved using CH_2_Cl_2_/pyridine/acetic anhydride (2/1/1 *v*/*v*/*v*) by stirring the mixture at room temperature for 30 min. After washing, the peptide was cleaved and deprotected using a solution containing 97% TFA, 3% triisopropylsilane (TIS) for 2 h. The resin was removed by filtration, and the filtrate was precipitated by the addition of diethyl ether. After centrifugation, the supernatant was removed, and the crude peptide was obtained, which was further purified by reverse-phase HPLC, followed by lyophilisation. The desired product was confirmed by matrix-assisted laser desorption/ionisation time-of-flight (MALDI-TOF) mass spectrometry.

### 2.2. Conjugation of CPP to Cytochrome C

Initially, equine cytochrome C (Sigma-Aldrich, Saint Louis, MO, USA) was solubilised in phosphate-buffered saline (pH 8.0) at a concentration of 0.83 mM. Subsequently, a 5-fold excess of the linker was dissolved in dimethyl sulfoxide (DMSO) and added to the cytochrome C solution at room temperature for a duration of 30 min. Without further purification, a 10-fold excess of purified cell-penetrating peptide (CPP) was dissolved in phosphate-buffered saline and added to the cytochrome C solution. The resulting mixture was incubated for 1 h at room temperature. Excess linkers and peptides were removed from the reaction mixture using a spin column with a molecular weight cut-off of 5000. The conjugation of CPP to cytochrome C was confirmed by liquid chromatography-mass spectrometry (LCMS).

### 2.3. Addition of Dimethylacetamide Anhydride (DMA)

A total of 50 equivalents of DMA were solubilised in dimethyl sulfoxide (DMSO) and reacted with **CPP-CytC** conjugates at a concentration of 0.83 µM. The reaction was carried out at 37 °C and pH 8.0 for a duration of 1 h. The excess DMA was subsequently removed from the reaction mixture using a spin column with a molecular weight cut-off of 5000. To prevent hydrolysis of DMA, the desired product **DMA.CPP-CytC** was freshly prepared right before the experiment and immediately utilised for each experiment.

### 2.4. Cell Culture

The HT29 human colorectal adenocarcinoma cells (ATCC, Manassas, VA, USA, no. HTB-38) were cultured in Roswell Park Memorial Institute (RPMI) 1640 Medium supplemented with fetal bovine serum (FBS) (10%) and penicillin-streptomycin (100 units/mL and 100 μg/mL, respectively). All cell culture experiments were conducted in a humidified atmosphere containing 5% CO_2_ at 37 °C. Different pH culture mediums were prepared by using HCl to decrease the pH of RPMI 1640 medium from pH 7.4 to pH 6.5. The pH was monitored throughout the experiment using a pH metre to ensure the pH was constantly maintained during the pH-responsive experiment.

### 2.5. Cellular Cytotoxicity Assay

HT29 cells were seeded at a density of 20,000 cells per well in a 96-well plate and cultured for 24 h. Subsequently, the culture medium was aspirated, and the cells were treated with cytochrome C conjugates in a serum-free medium. Following 24 h of incubation, the cytochrome C conjugates were removed, and the cells were washed with phosphate-buffered saline (PBS). A volume of 100 µL of 0.5 mg/mL MTT (3-(4,5-dimethylthiazol-2-yl)-2,5-diphenyltetrazolium bromide) tetrazolium solution was added to each well and incubated for 4 h at 37 °C. The MTT solution was then replaced with 50 µL of dimethyl sulfoxide (DMSO) for 10 min. The absorbance of each well was measured at 492 nm using a microplate reader.

### 2.6. Cell Apoptosis Assay

The Annexin V-FITC/PI double staining assay was conducted following the protocol provided by the Annexin V-FITC/PI Apoptosis Detection Kit BL110A (Biosharp, Hefei, China). Initially, HT29 cells were cultivated in RPMI 1640 medium supplemented with 10% FBS and penicillin-streptomycin (100 units/mL and 100 µg/mL, respectively), seeded at a density of 2.5 × 10^5^ cells per well in clear 24-well flat-bottom plates. The cells were allowed to adhere for 24 h at 37 °C in a humidified 5% CO_2_ atmosphere. Subsequently, they were treated with 40 µM cytochrome C conjugates in serum-free medium at pH 7.4 or 6.5 for another 24 h under the same conditions. Post-treatment, the cells were collected via trypsinization and centrifugation, and the pellet was resuspended in 50 µL binding buffer containing 2.5 µL Annexin V-FITC and 5 µL propidium iodide (PI). Following a 15-min staining at room temperature, the suspension was filtered through nylon mesh and diluted in 200 µL PBS for immediate analysis using a BD Accuri C6 Flow Cytometer, with data processed via Flowjo_V10 software.

### 2.7. In Vitro Imaging Experiment

HT29 cells were seeded at a density of 1 × 10^5^ cells per 35-mm confocal dish and incubated for 24 h. The culture medium was then replaced with 20 µM cytochrome C conjugates, and the cells were incubated for an additional 4 h. Following treatment, the cells were washed with phosphate-buffered saline (PBS) three times and stained with 1 µg mL^−1^ Hoechst 33,342 nuclear dye (Phygene) at 37 °C for 10 min. Cells were washed with PBS three times to remove excess dye. Cellular uptake of the cytochrome C conjugates was visualised using a Leica TCS SP8 high-speed confocal microscope equipped with solid-state lasers. The fluorescein was excited at 488 nm, and its fluorescence was monitored at 500–530 nm. The Hoechst 33,342 was excited at 405 nm, and its fluorescence was monitored at 415–480 nm. The images were digitised and analysed using Leica Application Suite X software.

### 2.8. HT29 Spheroid Culture and Confocal Microscopy

HT29 3D spheroids were obtained from a local company (BioArchitec Group Limited, Hong Kong, China) using Uni-Spheroid technology to sort a similar size (±10 µm) of 3D spheroid from a population of the spheroids. The spheroids were then grown in RPMI 1640 medium supplemented with 10% FBS and penicillin-streptomycin (100 units/mL and 100 µg/mL, respectively), at 37 °C in a 5% CO_2_ humidified atmosphere. All the spheroids were maintained at 37 °C in a 5% CO_2_-humidified atmosphere before the experiment. The spheroids were then treated with cytochrome C conjugates for 24 h. After treatment, the cells were stained with fluorescein isothiocyanate (FITC)-Annexin V and propidium iodide (PI) for 15 min and followed by the staining with Hoechst for 10 min at room temperature at room temperature before being taken for confocal microscopy.

### 2.9. 3D Spheroid Viability Assay

The 3D HT29 cell spheroids, obtained from Bioarchitec Group Limited, Hong Kong, China, were cultured in clear U-bottom 96-well plates. For recovery and attainment of a healthy state, the spheroids were incubated in RPMI 1640 medium, enriched with 10% FBS and penicillin-streptomycin (100 units/mL and 100 µg/mL, respectively), at 37 °C in a 5% CO_2_ humidified atmosphere for 4 h. Post-recovery, the medium was replaced with a serum-free medium, adjusted to pH 7.4 or 6.5, and treated with varying concentrations of DMA.CPP-CytC or CPP-CytC (ranging from 0 to 128 μM: 0 μM, 2 μM, 4 μM, 8 μM, 16 μM, 32 μM, 64 μM, and 128 μM). After a 24-h incubation period, the medium containing the reagents was discarded, and the spheroids were washed with PBS. Subsequently, 50 µL of CellTiter-Glo^®^ Reagent G7572 (Promega, Madison, WI, USA) was added to each well. Following a 30-min incubation at room temperature, the spheroids were transferred to a white flat-bottom 96-well plate, and the luminescence was measured using a Thermo Scientific Varioskan LUX Multimode Microplate Reader, facilitating the assessment of spheroid viability.

### 2.10. In Vivo Tumour Regression Experiment

HT29 cells, cultured in RPMI 1640 medium supplemented with 10% FBS and penicillin-streptomycin (100 units/mL and 100 µg/mL, respectively), were initially seeded in 15-cm cell culture dishes. The incubation was at 37 °C in a humidified atmosphere containing 5% CO_2_. Upon reaching 70% confluency, the cells underwent trypsinisation and were then centrifuged. The resulting pellet was resuspended in PBS. For in vivo studies, 1 × 10^6^ HT29 cells in 50 µL PBS were subcutaneously inoculated into the right side of dorsal flank region of mice. Once the tumours attained a size of 80–100 mm^3^, DMA.CPP-CytC (10 mg/mL, 0.6 mM, prefiltered through a 0.22 µm membrane) was administered intratumorally to the tumour-bearing mice. A PBS injection of identical volume served as the control. The mice’s body weight and tumour size were measured once every two days. Tumour dimensions were recorded using a digital micrometre calliper (SCITOP Systems), and volume (mm^3^) was calculated with the formula: tumour volume = (length × width^2^)/2. Monitoring of tumour sizes and body weight continued for 22 days post-treatment. Tumor volume data for the control group were collected from three mice. After 22 days, the mice were euthanised, and tumours were excised and preserved in 4% paraformaldehyde for subsequent histological analysis.

## 3. Results

### 3.1. Design and Synthesis of DMA.CPP-CytC

First, we modified the surface lysines on cytochrome C by using five equivalents of commercially available N-hydroxysuccinimide ester-maleimide (**NHS-MAL**) bifunctional linker in the presence of phosphate-buffered saline (PBS) pH 8.0 for 30 min to let the NHS react with the surface amine on the protein. Then, without further purification, 10 equivalents of chemically synthesised CPP (Appendix A) containing cysteine were added into the mixture for 1 h in the same buffer, followed by size exclusion spin column purification and LCMS characterisation (Figure 1). Our mass spectrometry data showed that the majority of the protein was conjugated with four CPP on the protein surface. Then, 50 equivalents of DMA were further added into the mixture for 1 h at room temperature and simple size exclusion filtration was performed to generate the **DMA.CPP-CytC** (Figure 2). Zeta potential of different molecules, including CPP, CPP-CytC, and DMA.CPP-CytC was measured to confirm the modification of the protein (Appendix A).

### 3.2. Internalisation of DMA.CPP-CytC

Upon synthesising **DMA.CPP-CytC**, we examined the translocation properties of the modified proteins using confocal microscopy at different culture pH values. As a comparison, we also synthesised unprotected CPP analogues (**CPP-CytC**). To enable visualisation under confocal microscopy, the N-terminus of the CPP on the protein was coupled with 5(6)-carboxyfluorescein (Appendix A). The experiment utilised HT29 human colorectal adenocarcinoma cells, cultured in two different pH conditions (pH 6.5 and pH 7.4), with the modified proteins added to each condition, followed by a 4 h incubation period. Confocal microscopy was used to monitor the fluorescent signal of the compounds inside the cells. Our findings revealed that in the absence of **DMA, CPP-CytC** translocated into the cytoplasm and nucleus of HT29 cells at both an acidic and neutral pH (Figure 2). Conversely, **DMA.CPP-CytC** demonstrated pH-dependent penetration into cells. An intense fluorescent signal was observed in both the cytoplasm and nucleus of HT29 cells at pH 6.5, whereas a significantly reduced fluorescent signal was observed at pH 7.4.

### 3.3. Cytotoxicity of DMA.CPP-CytC and Its Analogues at Different pHs

Considering the cellular apoptosis in the presence of cytosolic cytochrome C, the cell viability upon treatment with different cytochrome C analogues was further investigated. Initially, the cytotoxicities of native cytochrome C, **CPP-CytC**, and **DMA.CPP-CytC** were demonstrated. Our data showed that treatment with 40 µM **CPP-CytC** in HT29 cell culture resulted in extensive apoptosis at both pH 7.4 and pH 6.5. This evidence, in conjunction with the aforementioned confocal microscopy data, confirms that only **CPP-CytC** could lead to cellular cytotoxicity via internalisation into the cytoplasm, regardless of pH. Appendix A indicates that upon exposure to 40 µM native cytochrome C, linker-conjugated cytochrome C, and 400 µM of CPP peptide, HT29 cells did not exhibit significant cytotoxicity, confirming the cytotoxicity was caused by the function of cytochrome C.

Subsequently, a detailed dose-response curve for cell viability was constructed to determine the IC_50_ of the compounds at two different pHs. As shown in Figure 3A, **CPP-CytC** exhibited nonselective cancer-killing activity at both neutral and acidic pH levels. However, **DMA.CPP-CytC** demonstrated negligible cytotoxicity at pH 7.4 while exhibiting similar cytotoxicity to the non-DMA-protected **CPP-CytC** at pH 6.5. This confirms that the use of DMA to protect the CPP provides an acidic selectivity for protein delivery. Moreover, the preservation of the apoptotic effect confirms that the conjugation method does not significantly damage or denature the protein. Bovine serum albumin (BSA), used as a control protein and further modified by CPP and DMA, exhibited no cytotoxicity at the tested pH levels, thereby confirming the cytotoxicity of **DMA.CPP-CytC** stems from cytochrome C rather than peptide, linker, or DMA additions (Figure 3B).

In addition to cell viability, the apoptotic effects of cytochrome C analogues were determined via Annexin V and propidium iodide (PI) double staining. Initially, the unprotected **CPP-CytC** at pH 6.5 and 7.4 was investigated; quadrant analysis revealed that over 90% of the cells underwent apoptosis at both pH levels. These results indicate the nonselective internalisation of **CPP-CytC** into the intracellular space. For **DMA.CPP-CytC** at pH 6.5, a similar percentage of apoptotic cells was observed compared to **CPP-CytC**. (Figure 4) Under pH 7.4, a dramatic change occurred as the percentage of apoptosis was significantly reduced. This data set clearly demonstrated that DMA possesses a shielding effect for the CPP, preventing cytochrome C entry into the cells at a neutral pH.

### 3.4. Spheroid Cytotoxicity Assay and Internalisation

Tumour spheroids represent three-dimensional (3D) cell cultures that closely mimic the in vivo microenvironment of tumours, and they are frequently employed to model colorectal cancer using HT29 cells. As a result, we utilised the HT29 3D spheroid model to demonstrate the spheroid cytotoxicity effects of different cytochrome C analogues. CellTiter Glo (Promega, Madison, WI, USA) was used to determine the cell viability of the cell spheroid. As shown in Figure 5, our data suggest that **DMA.CPP-CytC** can selectively cause cell death in the spheroid at pH 6.5 but not at pH 7.4. Confocal images also showed that the PI and Annexin V positive signals were intensified in the low pH environment.

### 3.5. In Vivo Anticancer Effect of DMA.CPP-CytC

Ultimately, the anticancer activity of **DMA.CPP-CytC** was examined. Nude mice bearing HT29 xenografts were prepared, and upon the tumours reaching an approximate size of 80–100 mm^3^, **DMA.CPP-CytC** (10 mg/mL, 50 µL, 41.67 nmol) was administered intratumourally. An equivalent volume of PBS was injected intratumourally as a negative control. Following these treatments, tumour sizes were measured at two-day intervals for a total duration of 23 days. Additionally, the mice’s body weights were recorded, and no significant decreases in body weight were observed, indicating a lack of discernible toxicity (Figure 6).

## 4. Discussion

Cytochrome C is a water-soluble haemoprotein that is found in the mitochondria. The release of cytochrome C into the cytoplasm triggers the cell apoptosis process via cellular stress and caspase activation, ultimately leading to cell death [34,35]. Different research groups reported the delivery of bioactive cytochrome C via carriers such as liposomes and different types of nanoparticles [36,37,38,39,40,41]. Thus far, carrier-free delivery of cytochrome C remains rare, and nonspecific delivery is often encountered [42]. Recently, our group has developed several carrier-free strategies for selectively delivering bioactive proteins such as green fluorescent protein and ß-galactosidase, as well as small molecules, into tumour cells by conjugating them with tumour-specific binding peptides against tumour biomarkers such as integrin and EGFR [43,44,45,46,47]. These cargos can specifically target these surface biomarkers and enter intracellular space through receptor-mediated endocytosis. Although this approach is known for its specificity against different types of cancers, it is typically unsuitable for the majority of native proteins. This is because through receptor-mediated endocytosis, these proteins are trapped in the lysosome, and they are usually susceptible to degradation and denaturation in the low pH (<5.0) environment with the presence of different lysosomal enzymes inside the lysosome, and thus, losing their bioactivities [48]. CPP, while effective in preventing protein cargos from entering the lysosome, lacks the ability to differentiate between healthy and cancer cells. This nonselectivity poses a significant issue, as it can lead to unintended side effects in healthy cells, complicating the therapeutic application and potentially hindering its translatability into clinical practice [8].

As such, we aim to address the bottlenecks in bioactive protein delivery by employing a novel chemical strategy. This involves modifying the protein in conjunction with the DMA-CPP system, thereby engineering a new variant of the bioactive protein. Our approach is designed to enable selective penetration into the intracellular space of cancer cells, where the protein can effectively perform its native function. This innovative method not only enhances the delivery efficiency of therapeutic proteins but also ensures that their intrinsic functionalities are retained and effectively utilised within the targeted cellular environment. The DMA.CPP could selectively be deprotected in an acidic environment, and CPP could directly penetrate into the cytoplasm without going through the endosome or lysosome in an acid-specific manner. Furthermore, our carrier-free design, with its simplicity and directness, presents a potentially more translationally relevant and efficient therapeutic route. This is particularly advantageous in terms of dose control and consistency, addressing the batch-to-batch variation and intricate release kinetics often associated with carrier-based systems. As with all novel therapeutic agents and delivery systems, including nanoparticles, the immunogenic potential of our modified protein remains an area of exploration. This necessitates further detailed studies to comprehensively assess the immunogenicity and pharmacokinetics of the modified cytochrome C. Despite these unknowns, we are optimistic about the potential impact and translational feasibility of this strategy. Our preliminary findings suggest that it could represent a significant advancement in therapeutic delivery, warranting continued investigation and development. This direct approach balances the promise of innovation with the rigour of scientific validation, paving the way for a potentially new strategy in translational research and drug development.

## 5. Conclusions

In conclusion, our protein modification approach using a cell-penetrating peptide (CPP) combined with 2,3-dimethyl maleic anhydride (DMA) presents a promising strategy for intracellular protein delivery in a pH-dependent manner without using any nanocarriers. By using cytochrome C as a model protein and shielding the positively charged lysine residues of CPP with DMA, we successfully developed an acid-responsive protein delivery system. The effectiveness of this method was confirmed through various in vitro and preliminary in vivo assays, which showcased the apoptotic effect of the CPP-protein conjugate under acidic conditions. Our findings pave the way for the development of efficient protein delivery systems, particularly for cancer therapy applications.

## Data Availability

The data presented in this study are available on request from the corresponding authors.

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
