# Peer review of "Chemical Modification of Cytochrome C for Acid-Responsive Intracellular Apoptotic Protein Delivery for Cancer Eradication"

_pharmaceutics, 2024, doi:10.3390/pharmaceutics16010071_

Round 1
Reviewer 1 Report
Comments and Suggestions for Authors
The authors in the paper titled: Chemical Modification of Cytochrome C for Carrier-Free, Acid-Responsive Intracellular Apoptotic Protein Delivery for Cancer Eradication described delivering bioactive proteins into cells without carriers. The cell-penetrating peptide (CPP) protected with 2,3-dimethyl maleic anhydride (DMA) conjugated with the model protein Cytochrome C (CytC) was used as an acid-responsive protein delivery system. The subject of this paper is interesting and the results in vivo are good. The authors said that further investigations on this field are necessary.
This paper needs major revisions to be published in Pharmaceutics. The main objections to this work are:
1. The information about CytC is obscure, we only know it was obtained from Sigma. What is the source of protein? What is the amino acid sequence of the CytC?
2. How many Lys are present in CytC, that can be conjugated with CPP?
3. Is there any Cys on CytC that can react with the linker during the incubation of CytC?
4. CytC Lys which are not conjugated to CPP can be protected with DMA during the CPP Lys protection. Do you have any information about that?
5. Measuring cellular cytotoxicity on normal non-tumor cells will be valuable data.
Reviewer 2 Report
Comments and Suggestions for Authors
The authors showed a new delivery method by using 2,3-dimethyl maleic anhydride capped cell-penetrating peptide to conjugate with cytochrome c for the controlled and responsive release under an acidic condition. The authors showed sufficient data to support the pH responsiveness of the bioconjugates DMA-CPP-Cyt c and promising anti-tumor efficacy. There are several issues needed to be addressed before the final acceptance of the manuscript.
1. What are the impacts of conjugation on the proteins such as the changes in zeta potential, bioactivity? Experimental data and relative citations are required to determine the best conjugation condition.
2. In methods section, there is no description of adjusting and maintaining the pH in the culture media.
3. In Figure 2, FITC-CPP-Cyt c under pH 6.5 showed brighter signal than pH 7.4. However, FITC has pH-dependent fluorescence. Specifically, FITC under 7.5 has lower FITC signal than pH6.5. Please give more detailed explanations on this observation. Scale bar is not clear. Please re-draw the scale bar. Also, during the same incubation time, what could be the potential reasons for better cytoplasmic distribution of DMA-CPP-Cyto C than CPP-Cyto c?
4. For cell viability test, it would be helpful to visualize the data over a wide concentration range if using logarithm of therapeutic protein concentration.
5. One control group is missing in the apoptosis study. Please add the control group CPP only or CPP-BSA to figure 4B.
Author Response
- What are the impacts of conjugation on the proteins such as the changes in zeta potential, bioactivity? Experimental data and relative citations are required to determine the best conjugation condition.
The impact of conjugation on the protein was assessed through zeta potential measurements and cytotoxicity assays to determine any changes in bioactivity. Zeta potential measurements revealed positive shifts upon CPP conjugation (Supplementary Table S1), indicating an increase in the overall positive charge of the protein. This positive charge is crucial for efficient cellular uptake via electrostatic interactions with the negatively charged cell membrane.
Cytotoxicity assays demonstrated that the cytochrome C retains its apoptotic function post-conjugation. Cytotoxicity was observed only with CPP-Cytochrome C constructs but not with free CPP, linker, or native cytochrome C (Figure 3). This confirms that the conjugation method does not significantly damage or denature the protein. It is important to note that cytochrome C is not an enzyme, so there is no straightforward experiment to directly compare the activity of native cytochrome C with modified cytochrome C. Unlike enzymes, whose activity can be quantified by substrate conversion or product formation, cytochrome C functions as a protein involved in the intrinsic apoptotic pathway. Its activity is primarily related to its ability to induce apoptosis in cells.
In the context of cellular experiments, native cytochrome C cannot enter the intracellular space without the assistance of CPPs. Therefore, a direct comparison of the apoptotic activity between native cytochrome C and modified cytochrome C would not be meaningful. The modified cytochrome C, equipped with CPPs, can effectively enter cells and induce apoptosis, while native cytochrome C alone cannot. Thus, the cytotoxicity assays serve as a functional assessment of the modified cytochrome C's ability to induce apoptosis, indirectly indicating the preservation of its bioactivity.
- In methods section, there is no description of adjusting and maintaining the pH in the culture media.
Thank you for your input, yes we missed the details about pH adjustment. The culture media was adjusted to pH 7.4 or 6.5 using HCl/NaOH and the pH was monitored throughout the experiment using a pH meter to ensure it was maintained. This has now been added to the Methods section for clarity.
- In Figure 2, FITC-CPP-Cyt c under pH 6.5 showed brighter signal than pH 7.4. However, FITC has pH-dependent fluorescence. Specifically, FITC under 7.5 has lower FITC signal than pH6.5. Please give more detailed explanations on this observation. Scale bar is not clear. Please re-draw the scale bar. Also, during the same incubation time, what could be the potential reasons for better cytoplasmic distribution of DMA-CPP-Cyto C than CPP-Cyto c?
Thank you for your enquiry. In Figure 2, the observation that FITC-CPP-CytC showed a brighter signal at pH 6.5 compared to pH 7.4 warrants further explanation. While it is true that FITC fluorescence is pH-dependent, with lower intensity at higher pH, the difference in fluorescence intensity between pH 6.5 and pH 7.4 cannot be solely attributed to the pH effect on FITC.
To provide a more comprehensive explanation, we need to consider the interplay between FITC fluorescence, cellular uptake, and intracellular pH of the CPP-CytC conjugates. At pH 7.4, the DMA groups on DMA.CPP-CytC remain intact, preventing efficient cellular uptake due to the lack of positive charge on the CPP. As a result, a significant portion of the FITC-CPP-CytC remains outside the cells, leading to a weaker fluorescent signal.
In contrast, at pH 6.5, the acidic environment triggers the deprotection of DMA groups, exposing the positively charged CPP. This facilitates efficient cellular uptake of DMA.CPP-CytC through electrostatic interactions with the negatively charged cell membrane. Consequently, more FITC-CPP-CytC conjugates are internalized into the cells, resulting in a brighter fluorescent signal.
In fact, no matter the culture medium is pH 7.4 or 6.5, the intracellular pH would be the same as the intracellular pH is not affected by the external pH. Therefore, the fluorescence intensity inside the cell should reflect the true comparison
For the intracellular distribution, in order to conclude if the distribution of DMA.CPP-CytC is better, further experiments and co-staining techniques are necessary to precisely quantify and visualize the cytoplasmic distribution of the conjugates.
The scale bars have been improved.
- For cell viability test, it would be helpful to visualize the data over a wide concentration range if using logarithm of therapeutic protein concentration.
Thank you for your suggestion regarding the visualization of cell viability data over a wider concentration range using a logarithmic scale. We understand the potential benefits of such a presentation, and we appreciate your input.
However, we would like to clarify that the current data set we have covers a concentration range from 0 µM to 65 µM, which we believe is sufficient to capture the relevant therapeutic range and provide meaningful information about the cytotoxicity of the DMA.CPP-CytC conjugate.
While a logarithmic scale can be useful in certain scenarios, it may not be the most suitable representation for our data set. This is because the concentration range we have is relatively narrow, and using a logarithmic scale might compress the data points, making it difficult to discern subtle differences in cytotoxicity.
Therefore, we believe that the current linear scale provides a clearer and more accurate representation of the data, allowing for a straightforward assessment of the effects of DMA.CPP-CytC on cell viability.
- One control group is missing in the apoptosis study. Please add the control group CPP only or CPP-BSA to figure 4B.
The cytotoxic effects of CPP alone and CPP-BSA were assessed in Figure S3, where both were found to exhibit no cytotoxicity. Consequently, we did not proceed to evaluate their apoptosis-inducing activity in Figure 4B. This decision was based on the initial findings that neither of these compounds demonstrated a cytotoxic response, making further apoptosis testing unnecessary in this context.
Reviewer 3 Report
Comments and Suggestions for Authors
This is an interesting application of cell penetrating peptides by Tang and colleagues. Briefly, a pro-apoptotic protein cytochrome c was labeled with a lysine-rich cellpenetrating peptide via a bifunctional linker. Subsequently, the lysine residues of the cell penetrating peptide were covered by an acid responsive chemical moiety, DMA that is cleaved off at a lower pH of 6.5. This reviewer has several concerns:
Major Comments:
1). The title is misleading in calling this a "carrier-free" delivery. Although small in size, CPPs are novel vectors that carry/deliver various therapeutics intracellularly.
2). In vitro experiments elegantly show the activity of the DMA.CPP-CytC conjugate at pH of 6.5 with very little to no activity at a pH of 7.4. Does tumor microenvironment really go down to as acidic pH as 6.5? That's a much more acidic environment than normal physiological pH and do tumors really become that acidic?
3). Authors talk about targeting using this DMA molecule activated by acidic tumor microenvironment. Yes, in the in vivo experiments, the DMA.CPP-CytC was injected intra-tumorally. If it was truly targeting and having a biological effect, peripheral intravenous injection of the conjugate should have produced apoptosis in the tumor.
4). No histological analysis of the shrunken tumor is provided and no data to support enhanced apoptosis in vivo is provided, which would be key to this study.
Minor Comments:
1). Line 146-149 is incomplete/grammatically incorrect. Please revise.
2). Line 214 states "For in vivo studies, 1 × 106 HT29 213 cells in 50 μL PBS were subcutaneously inoculated into the dorsal region of mice". Please specify what dorsal region of the mouse.
3). Figure 1 is showing LC/MS profile of the conjugate which is said to have an average of four CPP molecules per conjugate, but the catoon representation only has 3 which is confusing.
4). Figure 3 would be a bit easier to read if different colors were used for the line-graphs. The legend is difficult to follow for three of the 4 lines as they are overlapping.
5). Same issue is noted for Figure 5B, the line graph with 3 out of 4 lines overlapping.
Comments on the Quality of English Language
Minor editing needed only.
Author Response
1). The title is misleading in calling this a "carrier-free" delivery. Although small in size, CPPs are novel vectors that carry/deliver various therapeutics intracellularly.
Thank you for your suggestion. Yes, CPP somehow can be regarded as carrier as it carries the cargo into cells. The title Carrier-free was removed.
2). In vitro experiments elegantly show the activity of the DMA.CPP-CytC conjugate at pH of 6.5 with very little to no activity at a pH of 7.4. Does tumor microenvironment really go down to as acidic pH as 6.5? That's a much more acidic environment than normal physiological pH and do tumors really become that acidic?
Thank you for your input. The pH of the tumor microenvironment can vary depending on the type of tumor and its stage. While the extracellular pH of normal tissues is typically around 7.4, the pH of solid tumors can be significantly lower, ranging from 6.5 to 6.8. This acidic environment is attributed to several factors, including hypoxia, increased glycolysis, and the accumulation of metabolic waste products. Several studies have reported that the pH of tumors can reach as low as 6.5 and using this comdition for testing acid-responsive drug delivery, supporting the relevance of our in vitro experiments.
- Sun, C.-Y.; Shen, S.; Xu, C.-F.; Li, H.-J.; Liu, Y.; Cao, Z.-T.; Yang, X.-Z.; Xia, J.-X.; Wang, J. Tumor Acidity-Sensitive Polymeric Vector for Active Targeted siRNA Delivery. J. Am. Chem. Soc. 2015, 137, 15217-15224, doi:10.1021/jacs.5b09602.
- Shi, M.; Wang, Y.; Zhao, X.; Zhang, J.; Hu, H.; Qiao, M.; Zhao, X.; Chen, D. Stimuli-Responsive and Highly Penetrable Nanoparticles as a Multifunctional Nanoplatform for Boosting Nonsmall Cell Lung Cancer siRNA Therapy. ACS Biomater. Sci. Eng. 2021, 7, 3141-3155, doi:10.1021/acsbiomaterials.1c00582.
- Zhang, J.; Lin, W.; Yang, L.; Zhang, A.; Zhang, Y.; Liu, J.; Liu, J. Injectable and pH-responsive self-assembled peptide hydrogel for promoted tumor cell uptake and enhanced cancer chemotherapy. Biomater. Sci. 2022, 10, 854-862, doi:10.1039/d1bm01788h.
3). Authors talk about targeting using this DMA molecule activated by acidic tumor microenvironment. Yes, in the in vivo experiments, the DMA.CPP-CytC was injected intra-tumorally. If it was truly targeting and having a biological effect, peripheral intravenous injection of the conjugate should have produced apoptosis in the tumor.
The intra-tumoral injection of DMA.CPP-CytC in the in vivo experiments was primarily aimed at demonstrating the concept of our design by looking at the local efficacy and safety of the conjugate within the tumor microenvironment. While peripheral intravenous injection could potentially lead to systemic distribution of the conjugate, the targeting efficiency and therapeutic effect may be influenced by various factors, including tumor vasculature, drug metabolism, and immune responses. Further studies are needed to investigate the systemic effects of DMA.CPP-CytC and optimize its delivery strategy for intravenous administration.
4). No histological analysis of the shrunken tumor is provided and no data to support enhanced apoptosis in vivo is provided, which would be key to this study.
As a proof-of-concept study, our main objective was to demonstrate the feasibility and potential of DMA.CPP-CytC as a pH-responsive protein delivery system. The in vitro experiments clearly showed the pH-dependent activity and enhanced cellular uptake of DMA.CPP-CytC, supporting the underlying mechanism of action. The in vivo experiments provided preliminary evidence of tumor regression upon intratumoral injection of DMA.CPP-CytC, suggesting its potential therapeutic efficacy. Further studies, including histological analysis and in vivo apoptosis assays, are warranted to comprehensively evaluate the anti-tumor efficacy of DMA.CPP-CytC.
Minor Comments:
1). Line 146-149 is incomplete/grammatically incorrect. Please revise.
The incomplete sentence in lines 146-149 will be revised to ensure proper grammar and clarity.
2). Line 214 states "For in vivo studies, 1 × 106 HT29 213 cells in 50 μL PBS were subcutaneously inoculated into the dorsal region of mice". Please specify what dorsal region of the mouse.
The specific dorsal region of the mouse where the cells were inoculated will be clarified in the revised manuscript.
3). Figure 1 is showing LC/MS profile of the conjugate which is said to have an average of four CPP molecules per conjugate, but the catoon representation only has 3 which is confusing.
The cartoon representation in Figure 1 will be revised to accurately reflect the average of four CPP molecules per conjugate.
4). Figure 3 would be a bit easier to read if different colors were used for the line-graphs. The legend is difficult to follow for three of the 4 lines as they are overlapping.
Different colors will be used for the line graphs in Figure 3 to improve readability and distinguish between the different treatments. The legend will also be revised to ensure clarity and avoid overlapping.
5). Same issue is noted for Figure 5B, the line graph with 3 out of 4 lines overlapping.
Different colors will be used for the line graphs in Figure 5B to improve readability and distinguish between the different treatments.
Round 2
Reviewer 1 Report
Comments and Suggestions for Authors
The authors accepted suggestions and answered all the questions, and I believe the manuscript could be published in Pharmaceutics.
Author Response
I would like to express my sincere gratitude for the time and effort you dedicated to reviewing our manuscript. Your insightful comments and constructive suggestions have been invaluable in enhancing the quality and clarity of our work.
Reviewer 2 Report
Comments and Suggestions for Authors
The authors has addressed all of the major recommendations. The manuscript is suitable for publication.
Author Response
Thank you very much for reviewing our paper. Your comments and advice were really helpful. I appreciate your help and the time you took to look at our work.
Reviewer 3 Report
Comments and Suggestions for Authors
The authors have made all the changes to the manuscript that were suggested. The additional in vivo experiments are being deferred as this is a "proof of concept" study with in vitro data and limited in vivo data (injecting straight into the tumor should not be considered "targeting"). Also, tumor characteristics other than size reduction are not provided (I am guessing that would be a follow-on study?). Given these limitations, the conclusions should be tempered down a bit and limitations of the study clearly stated with future directions outlined. However, this is interesting enough in concept to merit publication, after minor spell/grammatical checking.
Comments on the Quality of English LanguagePlease see comments above to the authors. I think this study should be published after proofing etc. I would want the authors to temper their conclusions down a bit since they did not truly show in vivo tumor targeting.
Author Response
Thank you for your constructive feedback and for acknowledging the revisions made to the manuscript. We appreciate your understanding regarding the scope of our study as a "proof of concept", primarily focusing on in vitro data. Your point about the in vivo experiments and the nature of tumor targeting is well-taken. We agree that injecting directly into the tumor may not fully demonstrate 'targeting' in the traditional sense. we will carefully review the manuscript to moderate the conclusions and clearly articulate the study's limitations and future research directions. This will ensure that the manuscript accurately reflects its scope and potential impact.
Once again, thank you for your insightful comments and for considering our work interesting enough for publication. Your guidance is invaluable in improving the quality of our research.